# SATS : Scenario-Anchored Topological Scoring in Figurative Expression

## Abstract

Figurative expressions remain challenging for language models, which often default to literal interpretations rather than capturing implicit meaning. This vulnerability affects the understanding of everyday dialogue and increases the exposure to adversarial prompts that exploit figurative or indirect phrasing. We integrate a topology-based algorithm into encoder-only architectures to strengthen signals relevant to figurative meaning and observe consistent improvements across multiple benchmarks. We further propose SATS, which achieves low latency and matches or exceeds most open-source LLMs while using 9.6× fewer parameters (within $0.8\%p$ of Qwen3). Our approach is lightweight and model-agnostic, and complements instruction-tuned LLMs by improving the robustness of detecting and interpreting figurative and implicit meaning.

## 1 Introduction

Figurative expressions are linguistic devices that enrich human conversation by conveying implied rather than surface meaning. They appear not only in poetry and fiction, but also in everyday conversation to evoke humor or convey reproach. For example, the sentence: "This young lady is 100% polyester" literally means that the woman is made of polyester, but its intended meaning is metaphorical: she behaves in a way that is unnatural or not humanlike. However, large language models (LLMs) often misinterpret such cases by taking surface cues like `polyester` literally. In Figure 1, given one scenario and five options, most models select option 1 or option 3 in which `polyester` appears verbatim. This behavior can be interpreted as a lexical-overlap bias with the scenario text. Figurative expressions affect not only task accuracy but also the safety of LLM services. Recent studies report that metaphorical expressions can be used to avoid hate speech detection (Zeng et al., 2025) or to gradually steer models toward harmful behavior via covert framing, commonly called jailbreak (Yan et al., 2025).

To mitigate these limitations, we propose a figurative expression detection algorithm that combines an encoder-only model with a topological approach. Our method achieves competitive detection performance with significantly fewer parameters than LLMs. It is designed to determine whether

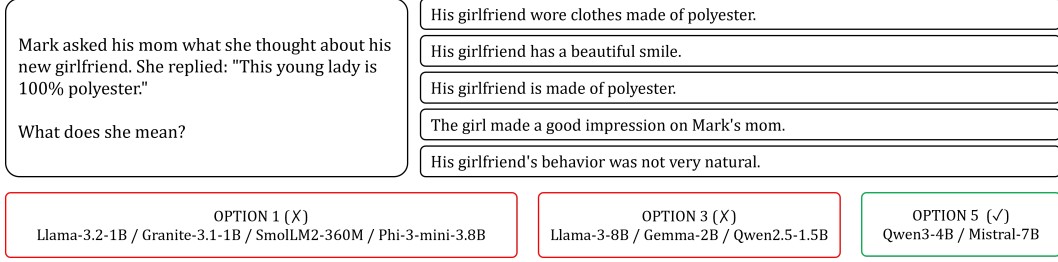

Figure 1: This example illustrates a Metaphor instance from the Pragmatics dataset (Hu et al., 2023). All LLMs are instruction-tuned models, and the temperature was fixed at 0.0 to ensure deterministic outputs. All models were evaluated with the same prompt, whose details are provided in Appendix E.

a sentence is figurative and, when candidate explanations are provided, to select the most plausible literal explanation. Our contributions are as follows:

- We introduce topological methods to representative figurative expression benchmarks, to the best of our knowledge for the first time, presenting a new paradigm beyond existing language-model approaches. On the Sarcasm benchmark, we observe an average improvement of about 1%p over BERT-based models. In FLUTE, using Representation Topology Divergence (RTD), we obtain at least 4.8%p accuracy improvement over LLMs with fewer than 3B parameters.

- We propose a multi-option algorithm, called Scenario-Anchored Topological Scoring (SATS), which directly compares persistent homology scores, overcoming pairwise-only comparison and high latency of RTD. In FLUTE, the proposed method reduces latency by approximately $275\times$ relative to RTD and improves accuracy by 1.9%p. Despite using about $9.6\times$ fewer parameters, it achieves higher accuracy than all eight LLMs except Qwen3. Even relative to Qwen3, the performance gap remains small at 0.8%p.

## 2 RELATED WORK

### 2.1 FIGURATIVE EXPRESSION

Figurative expressions pose a significant challenge for language models, as they require understanding beyond literal meaning (Liu et al., 2022). To address this, several detection approaches have been explored. For instance, Yang et al. (2024) proposed a GPT-based algorithm for verb metaphor detection, while Li et al. (2023) identified metaphors by contrasting the literal and contextual meanings of a word. The need for detection has also been emphasized in safety research. Zeng et al. (2025) and Yan et al. (2025) showed that LLMs fail to block metaphorical implicit hate speech and are vulnerable to adversarial jailbreaks. Although these studies mainly focus on safety and social risks, our work is distinct in that it investigates the structural characteristics of figurative expressions themselves through mathematical methodologies.

To support such research, several datasets have been introduced. Representative examples include the VUA (Leong et al., 2020) and FLUTE (Chakrabarty et al., 2022) datasets, which provides instances of Sarcasm, Simile, Metaphor, and Idiom in NLI format generated by GPT and human evaluators. In the sarcasm domain, SC V2 and MUStARD (Oraby et al., 2016; Castro et al., 2019) are widely used. Finally, Hu et al. (2023) investigates seven types of expressions, including Irony and Metaphor by requiring the selection of the most appropriate interpretation among multiple options.

### 2.2 TOPOLOGICAL DATA ANALYSIS

Early applications of persistent homology in NLP include Zhu (2013). Building on this direction, subsequent work constructs graphs from BERT attention and computes persistent homology features for spam detection and movie review sentiment classification (Perez & Reinauer, 2022). Other works reported meaningful performance improvements across benchmarks including Amazon Reviews, news classification, and grammatical acceptability by leveraging barcodes, mean of zero-persistent homology ($H_0M$), and RTD rules derived from attention graphs (Kushnareva et al., 2021; Cherniavskii et al., 2022). In addition, Proskurina et al. (2023) demonstrated strong performance on structurally different languages such as English and Russian using chordality.

Beyond NLP, Topological Data Analysis (TDA) has also been explored in other domains. Persformer proposed the first Transformer architecture capable of directly processing persistence diagrams, advancing representation learning and interpretability of TDA (Reinauer et al., 2022). Topoformer incorporated topographic organization into Transformer self-attention and demonstrated neuroscientific alignment (Binhuraib et al., 2024). More recently, TDA has been shown to enhance knowledge transfer in distillation (Kim et al., 2024) and provides a promising paradigm for graph and relational learning (Papamarkou et al., 2024).

Although TDA has become a significant research topic across NLP and beyond, no prior work has directly examined figurative expressions. This study addresses this gap by introducing a TDA-based approach to understanding figurative expression for the first time.

## 3 METHODOLOGY

This section provides a background of the topological approach and introduces two methodologies. First, it explains how existing methods are applied to figurative expressions and highlights their limitations. Then, it presents the algorithm proposed in this study to address these limitations.

### 3.1 BACKGROUND

In a Transformer, the attention matrix produced by each layer–head pair encodes pairwise token affinities and can be viewed as a weighted directed graph, with tokens as nodes and attention scores as edge weights. For an input of length $m$, let $W \in [0,1]^{m \times m}$ denote this matrix. For $\tau \in [0,1]$, retain only edges whose weights are at least $\tau$ to obtain a graph $G_\tau$. As $\tau$ increases, the family $\{G_\tau\}$ forms a filtration, a nested family of graphs under increasingly strict thresholds. Topological data analysis (TDA) summarizes how the topology of graph evolves along this filtration through persistent homology, which computes homology across the filtration and records birth–death pairs that mark the appearance and disappearance of features. These summaries are commonly visualized as barcodes, collections of intervals that encode feature lifetimes.

The Betti numbers vary with $\tau$: $\beta_0$ counts connected components and $\beta_1$ counts cycles. Barcodes record the birth and death of each component and cycle, where longer intervals indicate greater structural stability. In $H_0$, each component is born at $\tau = 0$ and dies at its merge time $d_i$, yielding intervals $[0, d_i]$. Under a standard similarity-to-distance transform and a decreasing-threshold filtration, the sum of $H_0$ barcode lengths is linearly related to the weight of a minimum spanning tree (Carlsson & Vejdemo-Johansson, 2022).

### 3.2 CLASSIFICATION WITH TOPOLOGICAL FEATURES

To validate a topological approach to figurative expressions, this study extracts attention matrices from BERT-based models and derives TDA vectors from them. These vectors are combined with the [CLS] token and used as input to a linear classifier for training and prediction. This approach is based on Kushnareva et al. (2021), and an example in our dataset is shown in Figure 2 in Appendix A.

The TDA vectors consist of three groups of features. Topological features include $\beta_0$ values, edge counts, and cycle counts across thresholds concatenated into a single vector. Barcode features include the sum and average lengths of the barcodes for $H_0$ and $H_1$, the number of intervals corresponding to the birth or death thresholds, and barcodes entropy. Finally, distance-to-patterns measures the normalized distance to predefined pattern graphs, reflecting how attention is distributed across specific tokens.

### 3.3 REPRESENTATION TOPOLOGY DIVERGENCE

Representation Topology Divergence (RTD) was first proposed in (Barannikov et al., 2022) as a metric that quantifies multi-scale topological differences between two point clouds obtained from the same sample set. In the NLP domain, Cherniavskii et al. (2022) converted sentence attention matrices into weighted graphs and applied RTD. In this formulation, RTD is defined according to the appearance of edges across thresholds: an edge that appears in only one graph is marked as birth, and an edge that appears in both graphs is marked as death. This process yields barcodes, with RTD computed as the sum of their lengths. Formally, for each bar $\alpha$ with birth and death interval $[\alpha_i, \alpha_j]$ in the barcode, its length is $\alpha_j - \alpha_i$. Thus, RTD is defined as

$$\text{RTD}(G_a, G_b) := \sum_\alpha (\alpha_j - \alpha_i),$$

where $G_a$ and $G_b$ are attention graphs. An example of RTD is detailed in Appendix B.

We adapt RTD under the hypothesis that figurative expressions are strongly associated with their correct interpretations. Given a figurative sentence with a literal explanation pair, we construct two inputs by concatenating the sentence with each explanation. We denote the sentence paired with the entailing explanation as $E$ and the sentence paired with the contradicting explanation as $C$. We then compute $\text{RTD}(G_E, G_C)$ and $\text{RTD}(G_C, G_E)$. Under our hypothesis, the correct explanation

yields more stable barcodes. Therefore, when $\text{RTD}(A, B) < \text{RTD}(B, A)$, we define the combination with $A$ as correct. To improve RTD performance, we employ a head ensemble strategy that, without additional parameter tuning, selects and aggregates layer–head combinations according to their validation performance on auxiliary data. Algorithmic details and procedures are provided in Appendix C.

Despite its utility, RTD has clear limitations. First, it is inherently restricted to pairwise comparisons and does not generalize to more than two candidates. Second, the computational complexity of RTD is difficult to analyze explicitly and has been empirically observed to incur a higher cost than $H_0 M$ (Cherniavskii et al., 2022). Similar concerns are noted in more recent work (Tulchinskii et al., 2025), which further motivates the need for scalable alternatives. To address these limitations, we propose a novel multi-hop approach in Section 3.4.

### 3.4 SCENARIO-ANCHORED TOPOLOGICAL SCORING (SATS)

Where appropriate, our exposition draws on Carlsson & Vejdemo-Johansson (2022) for definitions and notation. Because persistent homology is defined on simplicial complexes, we lift a graph to a simplicial complex. Let the thresholded graph be $G_\tau = (V, E_\tau)$ with the superlevel definition $E_\tau = \{\{a, b\} \in E : w_{ab} \geq \tau\}$, and let its clique (flag) complex be $K_\tau := \text{Cl}\,(G_\tau)$. Equipping the set of thresholds $I \subseteq [0, 1]$ with a descending total order $\succeq$, we obtain a filtration since $\tau_1 \succeq \tau_2$ implies $K_{\tau_1} \subseteq K_{\tau_2}$.

For each instance, we concatenate the scenario $S$ with each option $O_i$ to obtain a token sequence of length $T$ (the length may depend on $i$, but we keep the notation $T$). Let $A_{\ell,h}^{(i)} \in \mathbb{R}_{\geq 0}^{T \times T}$ denote the attention matrix of layer $\ell$ and head $h$, and mask special tokens in advance. For notational simplicity, we subsequently omit $(i, \ell, h)$. To remove asymmetry, set

$$U = \tfrac{1}{2}(A + A^\top), \qquad \text{diag}(U) = 0$$

and self-loops are removed. Because the dataset provides multiple options per scenario, we concatenate the scenario for each option. To focus on the scenario–option relation, we identify all scenario vertices into a single node $[S]$ through an equivalence relation $\sim$ and track connectivity on $K_\tau/\sim$.

Since we are concerned with connectivity only, we use $H_0$. Because $H_0$ is determined by the 1-skeleton, $K_\tau$ and $G_\tau$ are equivalent with respect to the connected components, and $H_0(K_\tau) \cong H_0(G_\tau)$ is true. Consequently, in practice, we compute on the graph $G_\tau$ without explicitly constructing the complex.

Define the death of an option vertex $o$ by

$$d(o) = \sup\{\tau \in I : o \text{ and } [S] \text{ are still disconnected in } K_\tau/\sim\},$$

which is attained as a maximum in a finite graph. Define the widest path by

$$\text{wp}(u, v) := \max_{p:\, u \rightsquigarrow v} \min_{e \in p} w_e, \qquad \text{wp}([S], o) := \max_{s \in S} p(s, o).$$

Then the following holds.

$$\forall\, o \in O: \quad d(o) = \text{wp}([S], o) = \max_{s \in S} \max_{p:\, s \rightsquigarrow o} \min_{e \in p} w_e.$$

Instead of directly evaluating the max–min rule of threshold scanning, we use the finite Katz index (Katz, 1953), which discounts and accumulates contributions of indirect paths:

$$\Phi_K(U; \beta) = \sum_{t=1}^{K} \beta^{t-1} U^t, \qquad \beta \in (0, 1),\ K \in \mathbb{N}.$$

We blend the multi-hop term with the direct term via

$$U_{\text{blend}} = \lambda U + (1 - \lambda)\,\Phi_K(U; \beta), \qquad \lambda \in [0, 1].$$

The scenario–option affinity is

$$u(o) := \max_{s \in S} (U_{\text{blend}})_{s,o}, \quad o \in O_i,$$

and the SATS score of option $i$ is

$$\text{SATS}(i) \ = \ \frac{1}{|O_i|} \sum_{o \in O_i} -\log\big(u(o) + \varepsilon\big), \qquad \varepsilon = 10^{-6}.$$

The final prediction is $\arg\min_i \text{SATS}(i)$, and the aggregation over scenario tokens used to compute $u(o)$ can be chosen as one of max, top-$k$, or softmax. The default hyperparameters are $K = 3$, $\beta = 0.6$, and $\lambda = 0.6$. We compute $u(o)$ using top-$k$ aggregation with $k = 3$. Formal proofs of the equivalence between death and the widest path and the stability of SATS are provided in Appendix D.

Each step costs $O(T^3)$ on a $T \times T$ matrix $U$ over $K$ steps, the total complexity is $O(K\,T^3)$. However, because our implementation executes all operations as dense matrix multiplications on the GPU via PyTorch, the measured latency was observed to be lower than that of RTD. The latency is primarily determined by $K$ and the token length $T$, while the hyperparameters $\beta$, $\lambda$, and the choice of aggregation (top-$k$/max/softmax) do not change the leading-order term.

## 4 Experiment

### 4.1 Datasets

Sarcasm Corpus V2 (SC V2) (Oraby et al., 2016) is a sentence-level sarcasm dataset with three types: GEN (generic situations), HYP (hyperbolic expressions), and RQ (rhetorical questions). In our experiments, we split each type into training (70%) and test (30%) sets. FLUTE (Chakrabarty et al., 2022) is an NLI-format figurative expression dataset constructed using GPT generation, expert annotation, and crowdsourced labeling. It comprises four categories: Sarcasm, Simile, Metaphor, and Idiom. In this study, we exclude Sarcasm and use the remaining three categories since the structure of the Sarcasm differs from the other categories. Because each item contains one hypothesis and two premises, we treat each hypothesis–premise pair as a single case. Consequently, the effective number of instances is half of the original. Pragmatics dataset (Hu et al., 2023) requires selecting the most appropriate sentence from multiple candidates for a given scenario. Each type contains 20–40 instances, and no training split is provided. We use the Irony and Metaphor types, and each scenario includes four or five candidate options. The overall composition of these datasets is summarized in Table 1.

### 4.2 Models

SC V2 employs BERT-based models, including bert-base-uncased (Devlin et al., 2018), roberta (base, large) (Liu et al., 2019a), and deberta-v3 (base, large) (He et al., 2021). The comparison models are fine-tuned on the training data, and in the TDA experiments, both the original and the fine-tuned models are compared.

In the FLUTE and Pragmatics experiments, we use models with fewer than 3B parameters and those with 3B or more. The small models include Granite-3.1-1B (Granite Team, 2024), Llama-3.2-1B (AI@Meta, 2024), SmolLM2-360M (Allal et al., 2025), Qwen2.5-1.5B (Team, 2024), and Gemma-2B. The large models include Mistral-7B-V0.3 (Jiang et al., 2023), Llama-3-8B, Qwen3-4B (Team, 2025), and Phi-3-mini-4k (Abdin et al., 2024). The FLUTE experiment additionally includes T5-Large (Roberts et al., 2022) and T5-FLUTE (Chakrabarty et al., 2022). All generative models, unless otherwise noted, are instruction-tuned versions.

For the RTD and SATS experiments, we use DeBERTa-v3-large, BART-large-MNLI (Lewis et al., 2020) and RoBERTa-large-MNLI (Liu et al., 2019b).

### 4.3 Setup

We run all experiments on four RTX 5090 GPUs (32 GB each). All evaluations are zero-shot on the test set except SC V2. All LLMs share the same prompt and decoding settings (`temperature`=0.0, `top_p`=1.0, `max_new_tokens`=2). Because we request only a numeric prediction from each LLM (a single integer), we cap `max_new_tokens` at 2. The full prompt, the negative-lexicon specification used to suppress tokens (e.g., "answer," "correct," "option"), and the constraint that forces generation of an option index are provided in Appendix E.

Table 1: Training and test set composition. Each cell reports the original count followed by the effective count (original / effective), values in parentheses indicate the number of options per scenario.

| | SC V2 | | | FLUTE | | | Pragmatics | |
|---|---|---|---|---|---|---|---|---|
| | GEN | HYP | RQ | Simile | Metaphor | Idiom | Irony | Metaphor |
| Train | 4564 | 814 | 1191 | 1250 / 625 | 1250 / 625 | 1768 / 884 | - | - |
| Test | 1956 | 350 | 511 | 250 / 125 | 248 / 124 | 250 / 125 | 25 (4) | 20 (5) |

Table 2: SC V2 results. Each dataset reports Accuracy and F1. Left columns show the Baseline (fine-tuned), and right columns show Topological Features (TDA, fine-tuned). All models used the same training settings batch size : 32 and epochs : 5. The best performance is highlighted in **bold**.

| | Overall | | | | GEN | | | | HYP | | | | RQ | | | |
|---|---|---|---|---|---|---|---|---|---|---|---|---|---|---|---|---|
| | Baseline | | TDA | | Baseline | | TDA | | Baseline | | TDA | | Baseline | | TDA | |
| Model | ACC | F1 | ACC | F1 | ACC | F1 | ACC | F1 | ACC | F1 | ACC | F1 | ACC | F1 | ACC | F1 |
| bert-base-uncased | 75.9 | 75.0 | 72.3 | 72.3 | 78.4 | 77.3 | 77.0 | 77.0 | 70.9 | 70.0 | 66.0 | 66.0 | 78.3 | 77.8 | 74.0 | 74.0 |
| RoBERTa-base | 79.2 | 79.3 | 78.1 | 78.7 | 83.9 | 83.6 | 80.4 | 80.9 | 73.4 | 74.4 | 73.1 | 74.0 | 80.4 | 79.8 | 80.8 | 81.3 |
| RoBERTa-large | 69.4 | 75.4 | 78.6 | 78.4 | 50.1 | 66.8 | 77.1 | 77.2 | 76.6 | 78.2 | 76.9 | 77.1 | 81.4 | 81.1 | 81.8 | 81.0 |
| DeBERTa-v3-base | 79.6 | 78.9 | 78.9 | 78.7 | 82.2 | 81.6 | 81.0 | 81.0 | 76.3 | 75.8 | 74.3 | 74.0 | 80.4 | 79.2 | 81.4 | 81.1 |
| DeBERTa-v3-large | 80.1 | 79.6 | **81.1** | **80.6** | 83.0 | 82.8 | 83.3 | 83.2 | 75.1 | 74.5 | **77.7** | 76.9 | 82.2 | 81.4 | **82.4** | **81.8** |

## 4.4 RESULTS

Table 2 reports the SC V2 results. Under identical training settings, models fine-tuned with TDA-based representations (right columns) achieved higher overall accuracy and F1 than models fine-tuned with standard methods only (left columns). Notably, DeBERTa-v3-large achieved the best overall performance, and consistent gains were also observed for RoBERTa-large. These observations suggest that cues related to sarcasm may already be encoded in attention, and that TDA-based representations expose this signal during training, yielding performance gains.

However, for base-scale models we observed degradations. A plausible explanation is that, compared to large models, they contain fewer layers and heads, which makes it harder to capture stable topological features. Although these models underperform relative to baselines counterparts, Table 6 in Appendix F shows that, even with frozen parameters, the topology-based score alone exceeds the chance level of $50\%$. Overall, the results indicate that language models internalize aspects of sarcasm and that topological methods can efficiently extract these signals and improve detection performance.

Table 3 summarizes the FLUTE results. With the exception of Qwen2.5, LLMs with at most 3B parameters perform only modestly above chance, while models with at least 3B parameters generally reach $85\%$ or higher, with Qwen3 leading. The topological baseline (RTD) peaks on the Simile subset when a head ensemble is used. It is competitive and even surpasses Mistral v0.3, but its overall accuracy remains below that of several LLMs, including Qwen3. Under SATS, both DeBERTa and RoBERTa obtain the best accuracy on Idiom. DeBERTa-v3-large also attains the highest accuracy on Simile and ranks second overall. Relative to Qwen3, it uses about $9.6\times$ fewer parameters while trailing by only 0.8%p, and it exceeds all other large LLMs. As shown in Table 4, SATS substantially outperforms RTD in speed: RTD is costly due to full-range threshold scanning, whereas SATS executes $O(K T^3)$ operations in parallel on GPUs using PyTorch, producing low latency. In summary, SATS delivers an accuracy comparable to or exceeding that of almost 10 times larger LLMs, while requiring far fewer resources and achieving lower latency.

We observe the same pattern on the Pragmatics dataset. As with FLUTE, all LLMs are evaluated under identical prompts and settings (details in Appendix E). Pragmatics comprises 25 four-choice questions for Irony (chance level $25\%$) and 20 five-choice questions for Metaphor (chance level $20\%$). Because the two subsets differ in both the number of questions and the number of options, the overall score is reported as a micro-average. Under micro-averaging, the overall chance level is $22.8\%$. Table 5 shows that most small LLMs operate at or below chance: Gemma is only marginally above chance on all tasks, and aside from Qwen2.5 other small models fall below chance. Qualitative analysis indicates a lexical-overlap bias: Low-performing models often choose options that reuse the salient words of the scenario rather than capturing scenario–option semantics. For instance, in the error cases of Figure 1, Options 1 and 3 repeat the token polyester and are selected most

Table 3: Experimental results (accuracy) for Simile, Metaphor, and Idiom in the FLUTE dataset. Italic text denotes experimental results cited from Chakrabarty et al. (2022). For each benchmark, the best performance is shown in **bold**, and the second-best performance is indicated by underlining.

| Model | Params | Overall | Simile | Metaphor | Idiom |
|---|---|---|---|---|---|
| **T5 baselines** | | | | | |
| flan-T5-large | 780M | 27.8 | 6.4 | 35.5 | 41.6 |
| *T5-flute* | 780M | *69.7* | *62.8* | *73.3* | *72.9* |
| **LLMs ≤ 3B params** | | | | | |
| Llama3.2-1B-Instruct | 1B | 51.4 | 44.0 | 57.3 | 52.8 |
| SmolLM2-360M-Instruct | 360M | 55.1 | 54.4 | 55.7 | 55.2 |
| Gemma-2B-Instruct | 2B | 56.7 | 43.2 | 72.6 | 54.4 |
| Granite-3.1-1B-Instruct | 1B | 60.1 | 48.4 | 71.8 | 60.0 |
| Qwen2.5-1.5B-Instruct | 1.5B | 83.7 | 70.4 | 89.5 | 91.2 |
| **LLMs ≥ 3B params** | | | | | |
| Mistral-7B-Instruct-v0.3 | 7B | 85.6 | 76.8 | 90.3 | 89.6 |
| Llama3-8B-Instruct | 8B | 89.1 | 78.4 | 94.4 | 94.4 |
| Phi3-mini-3.8B-Instruct | 3.8B | 89.1 | 78.4 | **96.0** | 92.8 |
| Qwen3-4B-Instruct | 4B | **91.2** | **81.6** | 95.2 | 96.8 |
| **MNLI-pre-trained baselines** | | | | | |
| RoBERTa-large-MNLI | 356M | 23.2 | 16.8 | 18.5 | 34.4 |
| BART-large-MNLI | 407M | 58.9 | 45.6 | 69.4 | 61.6 |
| **RTD Approach** | | | | | |
| BART-large-MNLI | 407M | 64.2 | 62.4 | 69.4 | 60.8 |
| BART-large-MNLI Head Ensemble | 407M | 76.7 | 74.4 | 72.6 | 81.6 |
| RoBERTa-large-MNLI | 356M | 76.1 | 67.2 | 83.1 | 84.3 |
| RoBERTa-large-MNLI Head Ensemble | 356M | 88.5 | **81.6** | 91.1 | 92.8 |
| **SATS Approach** | | | | | |
| DeBERTa-v3-large | 418M | 90.4 | **81.6** | 91.9 | **97.6** |
| RoBERTa-large-MNLI | 356M | 89.8 | 79.2 | 92.7 | **97.6** |

Table 4: FLUTE and Pragmatics latency in seconds by model group.

| | FLUTE | Pragmatics |
|---|---|---|
| **LLMs ≤ 3B params** | | |
| Llama3.2-1B-Instruct | 2.71 | 0.78 |
| SmolLM2-360M-Instruct | 3.33 | 0.63 |
| Granite-3.1-1B-Instruct | 8.19 | 1.53 |
| Gemma-2B-Instruct | 2.87 | 0.57 |
| Qwen2.5-1.5B-Instruct | 3.07 | 0.70 |
| **LLMs ≥ 3B params** | | |
| Mistral-7B-Instruct-v0.3 | 5.59 | 1.30 |
| Llama3-8B-Instruct | 6.34 | 1.42 |
| Phi3-mini-3.8B-Instruct | 3.71 | 0.91 |
| Qwen3-4B-Instruct | 4.72 | 1.04 |
| **RTD Method** | | |
| BART-large-MNLI | 596.12 | – |
| RoBERTa-large-MNLI | 1479.15 | – |
| **SATS Method** | | |
| DeBERTa-v3-large | 3.71 | 2.91 |
| RoBERTa-large-MNLI | 5.38 | 3.31 |

frequently. Among LLMs, Qwen3 achieved the highest performance, consistent with the FLUTE results. Under SATS, RoBERTa-large-MNLI achieved the best accuracy on Irony, and DeBERTa-v3-large ranked second on both tasks. In the Overall (micro) metric, RoBERTa-large-MNLI performed at a level similar to Qwen3, followed by DeBERTa-v3-large. Because the confidence intervals for

Table 5: This table reports accuracy on the Pragmatics dataset (Irony and Metaphor). Because the dataset is small and the numbers of test cases and options differ across subsets, the overall accuracy is computed via micro-averaging. We also report $95\%$ confidence intervals. For each benchmark, the best result is shown in **bold**, and the second-best result is underlined.

| Model | Overall (micro) | Irony | Metaphor |
|---|---|---|---|
| **LLMs $\leq$ 3B params** | | | |
| Llama3.2-1B-Instruct | 8.9 [2.5, 21.2] | 12.0 [2.5, 31.2] | 5.0 [0.1, 24.9] |
| SmolLM2-360M-Instruct | 8.9 [2.5, 21.2] | 12.0 [2.5, 31.2] | 5.0 [0.1, 24.9] |
| Granite-3.1-1B-Instruct | 15.6 [6.5, 29.5] | 24.0 [9.4, 45.1] | 5.0 [0.1, 24.9] |
| Gemma-2B-Instruct | 26.7 [14.6, 41.9] | 28.0 [12.1, 49.4] | 25.0 [8.7, 49.1] |
| Qwen2.5-1.5B-Instruct | 53.3 [37.9, 68.3] | 60.0 [38.7, 78.9] | 45.0 [23.1, 68.5] |
| **LLMs $\geq$ 3B params** | | | |
| Mistral-7B-Instruct-v0.3 | 57.8 [42.2, 72.3] | 52.0 [31.3, 72.2] | 65.0 [40.8, 84.6] |
| Llama3-8B-Instruct | 60.0 [44.3, 74.3] | 64.0 [42.5, 82.0] | 55.0 [31.5, 76.9] |
| Phi3-mini-3.8B-Instruct | 64.4 [48.8, 78.1] | 64.0 [42.5, 82.0] | 65.0 [40.8, 84.6] |
| Qwen3-4B-Instruct | **68.9** [53.4, 81.8] | 64.0 [42.5, 82.0] | **75.0** [50.9, 91.3] |
| **SATS Method** | | | |
| DeBERTa-v3-large | 66.7 [51.0, 80.0] | 68.0 [46.5, 85.1] | 65.0 [40.8, 84.6] |
| RoBERTa-large-MNLI | **68.9** [53.4, 81.8] | **80.0** [59.3, 93.2] | 55.0 [31.5, 76.9] |

large LLMs (including Qwen2.5) overlap substantially, we interpret SATS to provide performance comparable to that of LLMs. According to Table 4, taking into account the parameter scale and latency, SATS offers an efficient solution that reaches LLM-level accuracy with lower latency and much fewer resources, even in multiple choice settings.

## 5 DISCUSSION

### 5.1 ANALYSIS OF SELECTED ATTENTION

Figure 5 summarizes the distribution of high-accuracy layer–head pairs in RoBERTa-large-MNLI and DeBERTa-v3-large under SATS. The color legend is as follows: yellow marks the best pair on FLUTE, green marks the best pair on Pragmatics, and orange marks the top 3 (duplicates allowed) layer–head pairs per dataset. For visualization, pairs are drawn under the same test conditions in the experiments. The procedure uses fixed tokenizers and models without stochastic components, so variability due to random seeds is negligible. For RoBERTa, aside from $(1, 11)$, $(3, 15)$, high-performing pairs tend to cluster in the mid-to-late layers. DeBERTa shows a similar tendency, with some early layers also appearing among the top-3. Examining the layer–head distribution selected by the RTD head ensemble, BART exhibits a relatively even spread across layers, whereas RoBERTa, while including some early layers, concentrates more frequently in mid-to-late layers. These observations align with previous findings that semantic information is more prominent in later layers of RoBERTa models (Li et al., 2021). Details on the selected attention heads in RTD are reported in Appendix G.

### 5.2 OPERATION OF SATS METHOD

Figure 7 (A) visualizes how SATS selects the correct answer on the FLUTE Simile dataset for DeBERTa-v3-large. Each token is obtained by encoding the input with the tokenizer of the model. Special tokens were removed in advance, whereas punctuation marks, including apostrophes, were retained. Here, $\tau \in [0, 1]$ denotes the threshold on edge weights in the multi-hop $\Phi$. As $\tau$ decreases from 1 to 0, higher-weight edges are activated in descending order, and newly activated edges are highlighted in red. When connections appear at relatively large values of $\tau$, the $u(o)$ of the corresponding scenario–option pair tends to increase. Consequently, the SATS value decreases, making the option more likely to be selected as correct. In this example, Option 2 acquires edges earliest and ultimately attains the smallest SATS value.

The bar chart on the right displays $u(o)$ for each option token and larger values indicate stronger associations with the scenario tokens. For Option 1, $s_1$ and $o_1$, both `gesture`, connect first, but the $u(o)$ for the core token `awkward` is the lowest, suggesting that the meaning of the option does not align with the scenario. By contrast, in Option 2, $s_5$ (`ballet`) and $o_3$ (`elegant`) connect first, supporting that the implied meaning of the scenario, `ballet` like movements are `elegant`, is captured. Similarly to Option 1, the $s_1$ and $o_1$, `gesture`, link forms at the next threshold. Despite sharing the same lexical form, it emerges later than the (`ballet`, `elegant`) pair, indicating that SATS prioritizes contextually coherent connection patterns over simple lexical overlap. (B) and additional examples are provided in Appendix I.

## 6 CONCLUSION

This study shows that a topological approach to figurative expressions can effectively capture metaphorical meaning within language models. In particular, the proposed SATS extends naturally from binary labels to multiple choice settings, achieving higher accuracy and lower latency than RTD. SATS captures contextually coherent semantic connections between scenarios and options without relying on simple lexical overlap, and it strengthens scenario–option coupling via multi-hop propagation. Furthermore, SATS, implemented with an encoder-based model that uses far fewer parameters than large LLMs, achieves accuracy comparable to or exceeding that of LLMs on multiple benchmarks. Given its reduced resource requirements, the SATS pipeline can serve as a practical component for integration with instruction-tuned LLMs to assist in detecting and understanding figurative expressions.

## 7 LIMITATIONS AND FUTURE WORK

The SATS approach has several limitations. Although parameter-efficient, its accuracy remains approximately 2 percentage points lower than that of Qwen3. Because our experiments adopt a classification setting with an encoder-only model, the method does not produce descriptive outputs when no literal interpretation is provided. In addition, the available evaluation sets are also small, FLUTE includes more than 100 test cases and thus supports some degree of generalization, whereas Fine-Grained Pragmatics provides only 20–25 cases per type, so the resulting estimates carry high variance.

Future research directions to address these limitations are as follows. First, we aim to narrow the performance gap of SATS by fine-tuning the base encoder to specialize in figurative expressions. On SC V2, fine-tuning strengthened the topological features and improved accuracy, which provides empirical support for this direction. As ongoing work, we are investigating the integration of SATS upstream of instruction-tuned LLMs as a lightweight preprocessing module. When figurative language is detected, the LLM generates multiple literal-interpretation candidates. SATS then selects the most plausible candidate using its multi-option scoring and supplies it as auxiliary input to the base model and to the policy filter. A comprehensive evaluation of safety is left to future work. Second, we plan to broaden benchmarks for the evaluation of generalization. For datasets without labels of literal interpretation (e.g., HYPO, LCC, TroFi Badathala et al. (2023)), we will generate plausible candidates of literal interpretation via LLM-based data augmentation and expand the evaluation set through human verification, enabling more precise measurement across diverse domains.

**LLM Usage** We used large language models only as assistive tools for (i) polishing and stylistic refinement of the manuscript, and (ii) surfacing candidate papers during literature search. All substantive content (problem formulation, methods, experiments, analyses, and conclusions) was written by the authors. Any text or citations suggested by an LLM were reviewed and verified by the authors, and we cite only sources we personally inspected. No non-public data or reviewer materials were uploaded to third-party services, and we preserved double-blind anonymity.

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
