# OpenReview forum: "SATS : Scenario-Anchored Topological Scoring in Figurative Expression"
_ICLR.cc/2026/Conference — ICLR 2026 Conference Withdrawn Submission_

### Official Review · Reviewer_mbxv · 2025-10-31

**Soundness:** 2
**Presentation:** 2
**Contribution:** 2
**Rating:** 2
**Confidence:** 3

**Summary:**

The paper introduces a new method for figurative language understanding based on topological data analysis. The approach complements existing LLMs by increasing their robustness and achieves performance gains on fig lang understanding benchmarks.

**Strengths:**

- Paper introduces a new method (SATS). I do not feel qualified to review the method's validity due to lack of background in topological data analysis, but it appears to be novel and sound.
- The paper benchmarks a variety of LLMs, both masked (roberta, bert, deberta) and autoregressive (qwen, llama, mistral etc.). The method yields comparable performance to larger models or less efficient methods.

**Weaknesses:**

- *Motivation*: the method presented seems very general, but the application is only to a narrow domain of figurative expressions. It is unclear why this domain was chosen to test this method. Is there something about topology of LLM representations that is inherently useful for figurative language understanding?
- *Presentation*: the method presentation is extremely dense and technical, and without the relevant background it is very hard to parse and understand the motivation and intuition behind it. There is almost no description of how the method builds upon or is different from other work in TDA: for example, limitations of RTD are mentioned in 3.3 but *how* does the new method address these limitations is not properly explained. Perhaps the paper would be a better fit to a more niche community, such as in a targeted conference or workshop.
- *Statistical validity*: the effect of the method is very small. In Table 2, F1 is only better sporadically, by extremely small margins. Since no confidence intervals or even just standard deviation are reported, no p values or statistical tests conducted, these results may very well stem from simple statistical noise. The generative models appear not to be fine-tuned which is not a very fair comparison to the method that appears to be trained on the train set. Overall, while efficiency gains are impressive, it appears performance gains are dubious and thus question whether such a complex method is necessary for this task.
- *Discussion*: The discussion section could be more detailed and provide more examples and intuitive explanations. It was not clear how this method in particular is advantageous for figurative meaning. Many figures are referred to Appendix but should really be part of the main paper to explain the point better.

**Questions:**

-

---

### Official Review · Reviewer_26Ua · 2025-11-01

**Soundness:** 3
**Presentation:** 2
**Contribution:** 2
**Rating:** 4
**Confidence:** 3

**Summary:**

This paper targets the weaknesses of large language models (LLMs) in handling figurative expressions by introducing the Scenario-Anchored Topological Scoring (SATS) technique. SATS is a lightweight, largely model-agnostic approach that utilizes topological properties of attention to select the best literal interpretation of a figurative expression out of several candidates. Empirically, it achieves comparable performance to much larger language models (400M vs 8B parameters) on figurative language benchmarks while running at a far lower latency than a naive relative topology distance (RTD) approach. The main conceptual contribution is a demonstration that the topology of attention within encoder-only language models implicitly carries information relevant to figurative expressions, with SATS serving as a concrete implementation of that claim.

**Strengths:**

S1: The SATS algorithm is mathematically well described and represents a thoughtful extension of previous topological analyses. While the technique has limitations (as authors acknowledge), through this and their precursor findings, they convincingly demonstrate that the information necessary to discern figurative language is encoded within the attention mechanism of encoder-only models.

**Weaknesses:**

W1: The paper is densely written and challenging to follow, with only a single figure in the main text. Although the authors logically build toward SATS by introducing precursor methods, the presentation could be improved by briefly previewing or summarizing SATS earlier (such as after Section 3.1) or by providing an intuitive explanation at the start of section 3.4. As it currently stands, the explanation of SATS seems somewhat disconnected from the account of RTD.

W2: From an organizational standpoint, some appendix figures referenced in the main text (such as Figures 5 and 7) would improve readability and understanding if included in the main text, possibly in an abridged form. Conversely, results such as latency as reported in Table 4 seem to be better suited for the appendix.

W3: Finally, and most importantly, while the performance of SATS is impressive given the parameter count, the issue of SATS requiring literal interpretations poses a significant limitation for its practical applicability (though the authors present a possible solution in lines 466-470). As the introduction and related work emphasize, detecting figurative language within longer texts remains a challenging open problem. The results in Table 3 would suggest that larger models are generally more successful at detecting such figurative expressions, as smaller models often struggle on the included benchmarks. In the authors’ proposed solution, following detection, these larger models would also then generate the literal interpretation candidates that SATS then scores. However, it would seem that the large models themselves are already capable of correctly interpreting the figurative language directly, as per Table 3, and so the incremental practical value of adding SATS on top may be limited.

**Questions:**

Q1: Is there any insight on how well SATS performs using the information from the base models? The authors mention that TDA outperforms chance on such models, it would be interesting to see if such is the case for SATS as well.

Q2: While outside of the scope of this paper, could SATS be adapted to decoder-only models, despite their causal attention masking?

---

### Official Review · Reviewer_tCAG · 2025-11-01

**Soundness:** 1
**Presentation:** 3
**Contribution:** 1
**Rating:** 0
**Confidence:** 5

**Summary:**

This paper argues that language models are not good at figurative expressions and proposes a figurative language detection algorithm to handle these cases. The paper proposes a method to upweight certain signals in the sentence to improve on figurative language benchmarks.

**Strengths:**

- The paper is well-written and easy to read.
- The method is clever and has good results.

**Weaknesses:**

The premise of the paper is flawed, as the LLMs evaluated are all very weak. Any frontier LLM is able to understand figurative language. For example,
- ChatGPT: https://chatgpt.com/share/6905a204-5be4-8011-82f4-4e5196585f3f
- Gemini: https://gemini.google.com/share/95e8e239876f
- Grok: https://grok.com/share/c2hhcmQtNQ%3D%3D_b61a5409-42de-4e0b-9d4c-64cdee27caee

It is thus unclear what the usefulness of the paper is.

**Questions:**

What are the evaluation results on these benchmarks for GPT 5 (minimal thinking), Claude Sonnet 4.5, Gemini 2.5 Flash, and Grok 4?

---

### Official Review · Reviewer_R5mt · 2025-11-01

**Soundness:** 2
**Presentation:** 3
**Contribution:** 2
**Rating:** 2
**Confidence:** 3

**Summary:**

The paper proposes a method that converts an encoder’s attention into a lightweight graph and uses a truncated multi-hop connectivity measure to pick the correct option in figurative-language multiple-choice tasks. It positions the method as a faster, simpler alternative to representation-tension diagnostics, and compares against both RTD and small LLMs.

**Strengths:**

Clear problem framing around figurative-language; method is simple and efficient.
The paper is well written.

**Weaknesses:**

The authors claim that the method combined with encoder models achieves accuracy comparable (or exceeding) to that of decoder LMs. However, the evaluation of LMs is constrained and misaligned with the model’s intended capabilities. The evaluation involves instruction-tuned variants of LMs, and works by generating up to 2 tokens with a shared prompt across the models. These models were trained with a prompt template unique to each model family, and in most cases were trained to generate CoT before providing the final answer. A fair evaluation of Instruct models would involve wrapping the shared user prompt within model-specific template, and let the model generate the CoT before providing the final answer, which would be later extracted.
For more simplistic evaluation, one would take the _base_ variants of decoder LMs, and prompt them with in-context examples to induce a specific pattern of behaviour.
Moreover, the encoder models were _trained_ before evaluation, and for a comprehensive comparison a similar setting could be applied to _base_ decoder LMs, e.g. by using parameter-efficient finetuning methods to lower the number of trainable parameters.

Additionally, the scope of the work is fairly narrow and tied to a specific, linguistically constrained task setup. The method primarily targets figurative and idiomatic multiple-choice benchmarks, which limits its practical relevance for broader tasks.

**Questions:**

1. During evaluation of Instruct language models, is a correct prompt template applied to each of them?
2. As mentioned earlier, the evaluation of LMs is misaligned. For instruct models it should prompt them with correct prompt template, and let them generate CoT before providing final answer. For base models, they should be prompted with a few in-context examples in a chosen format.

---

### Note · Authors · 2025-11-13

I have read and agree with the venue's withdrawal policy on behalf of myself and my co-authors.